# In Vitro Preformed Biofilms of *Bacillus safensis* Inhibit the Adhesion and Subsequent Development of *Listeria monocytogenes* on Stainless-Steel Surfaces

**DOI:** 10.3390/biom11030475

**Published:** 2021-03-22

**Authors:** Anne-Sophie Hascoët, Carolina Ripolles-Avila, Brayan R. H. Cervantes-Huamán, José Juan Rodríguez-Jerez

**Affiliations:** Human Nutrition and Food Science Area, Departament de Ciència Animal i dels Aliments, Universitat Autònoma de Barcelona (UAB), Edifici V-Campus de la UAB, 08193 Bellaterra (Cerdanyola del Vallès), Barcelona, Spain; annesophiehascoet@hotmail.com (A.-S.H.); carolina.ripolles@uab.cat (C.R.-A.); brayancervanteshuaman@gmail.com (B.R.H.C.-H.)

**Keywords:** biofilms, *Listeria monocytogenes*, pathogen inhibition, microbiota, surfaces, *Bacillus safensis*

## Abstract

*Listeria monocytogenes* continues to be one of the most important public health challenges for the meat sector. Many attempts have been made to establish the most efficient cleaning and disinfection protocols, but there is still the need for the sector to develop plans with different lines of action. In this regard, an interesting strategy could be based on the control of this type of foodborne pathogen through the resident microbiota naturally established on the surfaces. A potential inhibitor, *Bacillus safensis,* was found in a previous study that screened the interaction between the resident microbiota and *L. monocytogenes* in an Iberian pig processing plant. The aim of the present study was to evaluate the effect of preformed biofilms of *Bacillus safensis* on the adhesion and implantation of 22 strains of *L. monocytogenes*. Mature preformed *B. safensis* biofilms can inhibit adhesion and the biofilm formation of multiple *L. monocytogenes* strains, eliminating the pathogen by a currently unidentified mechanism. Due to the non-enterotoxigenic properties of *B. safensis*, its presence on certain meat industry surfaces should be favored and it could represent a new way to fight against the persistence of *L. monocytogenes* in accordance with other bacterial inhibitors and hygiene operations.

## 1. Introduction

*Listeria monocytogenes* remains one of the most important public health challenges for many food industries [1,2], including those in the meat sector [3]. In 2017, a total of 2502 confirmed listeriosis cases were reported in the European Union/European Economic Area [4]. Considering the data from previous years, the European Food Safety Authority (EFSA) and European Centre for Disease Prevention and Control (ECDC) has detected a constant increasing tendency of outbreaks since 2008 [4,5]. Moreover, listeriosis presents a high proportion of cases needing hospitalization in susceptible people [5]. Found in both food production and retail outlets, its ability to maintain itself in adverse environmental conditions makes it a very persistent pathogen that is greatly feared by the sector [1,2]. It is often associated with areas that have undergone an incorrect cleaning and/or wet process such as sinks or other areas that are difficult to access. Furthermore, many strains of *L. monocytogenes* can adhere to different surfaces and form biofilms. This foodborne pathogen can tolerate adverse conditions such as exposure to disinfectants, forming these structures or deploying mechanisms of resistance to stress. Therefore, *L. monocytogenes* survive for long periods in food processing plants [6,7,8], implying a permanent risk of cross-contamination of the products [6]. Bagge-Ravn et al. [9] and Bagge-Ravn et al. [10] isolated an identical *L. monocytogenes* strain, using the technique of random amplification of polymorphic DNA, which had been registered in 1995 in the same smokehouse. Ortiz et al. [7] revealed the persistence of different strains of this pathogen for three years in the same industry.

Various physical methods including hot steam, UV light, ultrasounds, and chemical strategies involving sodium hypochlorite, sodium hydroxide solutions, hydrogen peroxide, and peracetic acid, among others, have been used to control the presence of biofilms in the food industry, both inside pipes and on work surfaces [11,12]. The proposed approach for the microbiological control of food contact surfaces has traditionally been to maximize the reduction of microbial load. Accordingly, products and strategies to maximize cleaning and disinfection operations have been designed. However, given that they do not eliminate persisting *L. monocytogenes* biofilms, novel strategies such as bacteriophages, bacteriocins, quorum sensing (QS) inhibitors, antimicrobial peptides (AMPs), and essential oils have been investigated [11,13]. Another interesting strategy is based on the control of pathogens such as *L. monocytogenes* through the resident microbiota on surfaces. These resident species can inhibit the adhesion and proliferation of pathogens or, contrarily, favor their implantation and development in the form of mixed biofilms [6]. Analyzing and manipulating the profile of the resident microbiota (i.e., potentiating inhibitors and eliminating bacteria that promote the growth and/or survival of this pathogen) may be an essential new approach to combat *L. monocytogenes* biofilms [14]. The resident microbiota can have a strong effect on the probability of finding *L. monocytogenes* in food facilities [15]. A recent study proposed that the surface hygiene in the food industry could be reconsidered through the use of this type of microorganism, given they do not show any deterioration effect on food products [16]. The current problem of antimicrobial resistance (sensu lato) to antibiotics and disinfectants highlights the need to look for alternatives. Lactic acid bacteria have aroused much interest as a biocontrol agent on surfaces and packaging and as bioconservatives in the food itself [17,18,19,20,21]. However, there are other genera of interest naturally present on surfaces in food industries. Therefore, the exploration of species belonging to the resident microbiota of the food industries is not only an interesting ecological alternative to study, but it also opens up a field of study with great prospects for the future. This strategy would also be in line with the new consumer trends that increasingly reject the use of chemical substances in the food sector in general. Consumers associate more food risks with the presence of chemical products, pesticides and/or residues in food. In the “Global Consumer Food Safety and Quality” study conducted by TraceOne, more than 3000 consumers from nine different countries, including Spain, were interviewed [22]. In general, consumers also associate the absence of chemical substances with improved health [23]. Society fears being exposed to chemicals and their residues in food without their knowledge [24]. Consequently, resorting to the natural microbiota present in the food industries could constitute a complementary control strategy that is more acceptable to consumers.

One microorganism was selected for showing potential inhibition to the target foodborne pathogen from a first study by Hascoët et al. [14]. This previous investigation evaluated the resident microbiota on the surfaces of an Iberian pig processing plant and screened microorganisms that have an interesting interaction with *L. monocytogenes* [14]. This microorganism was *B. safensis*, which is a soil bacterium found in many habitats and capable of surviving in hostile environments [25,26]. Several studies indicate that *B. safensis* has many interesting properties and the potential for various industrial applications due to the production of enzymes and secondary metabolites of industrial interest [25]. This microorganism can produce biosurfactant substances with lipopeptidic structures, such as surfactin. This compound has been shown to have antibacterial properties against multiple pathogens. Specifically, it has showed anti-planktonic activity and anti-adhesion activity of more than 80% for *Staphylococcus epidermidis* biofilms [27]. In another study, it was shown that *B. safensis* can generate another biosurfactant with a structure very similar to pumilacidin [28]. Regarding the production of enzymes, *B. safensis* can produce amylases, lipases, proteases, β-galactosidase, and cellulases, among others. Another interesting finding is that *B. safensis* is capable of blocking virulence factors and the formation of biofilms of *Cryptococcus neoformans* and *Candida albicans*. The antipathogenic mechanism seems to be specific and based on the targeting of fungal cell wall [26].

The general objective was to evaluate the effect of preformed biofilms of *Bacillus safensis,* the potential inhibitor, on the adhesion and subsequent development of biofilms of *L. monocytogenes*. Four experimental studies were proposed, which sought to develop the following specific objectives: (i) verification of the ability of *B. safensis* to form biofilms, (ii) evaluation of the effect of preformed biofilms of *B. safensis* on 22 strains of *L. monocytogenes*, (iii) validation of the previous experimental design assessing the effect of preformed biofilms of *Pseudomonas* spp. and *C. zeylanoides*, and (iv) characterization of the effect of the biofilms of *B. safensis* on *L. monocytogenes* in order to approximate the mechanism of action.

## 2. Materials and Methods

### 2.1. Evaluation of the Ability of B. safensis to Form Biofilms

#### 2.1.1. Study Surface

Grade 2B AISI 316 stainless-steel coupons (Premium Lab, Barcelona, Spain) (2 cm in diameter and 1 mm thick) were used in the four experimental studies, prior to which they were subjected to cleaning and disinfection procedures. The stainless-steel coupons were first cleaned with a non-bactericidal detergent (ADIS Higiene, Madrid, Spain), then disinfected with 70% isopropanol (Panreac Química, Castellar del Vallès, Spain) and dried in a laminar flow cabinet (PV-30/70, Telstar, Terrassa, Spain). This procedure was carried out following the European standard UNE-EN 13697:2015, related to non-porous materials [29]. The stainless-steel coupons were previously autoclaved for 15 min at 121 °C to guarantee sterility.

#### 2.1.2. Inoculum Preparation

The isolated and identified strains from the ecological study carried out on the Iberian pig processing plant [14] were lyophilized. Prior to use, the lyophiles were rehydrated with 9 mL of Tryptic Soy Broth (TSB; BioMérieux, Marcy l’Etoile, France) and incubated at 30 °C for 40 to 48 h. Once the incubation time had elapsed, they were transferred to Tryptic Soy (TSA; Oxoid, Madrid, Spain) plates and incubated at 37 °C for 18 to 24 h. Isolated colonies were further transferred to inclined TSA tubes, which were incubated at 37 °C for 24 h and stored at a temperature of 4 °C, keeping them for a maximum period of one month. The microorganisms were reactivated by inoculating them on TSA plates and incubating them at 37 °C for 18 to 24 h. Bacterial inoculums were prepared from these cultures to form the biofilms. *B. safensis* inoculum was prepared by transferring multiple isolated colonies into TSB until reaching a turbidity of 0.1 McFarland units, equivalent to approximately 10^6^ CFU/mL. The bacterial suspension was diluted decimally in 9 mL peptone water tubes (BioMérieux, Marcy l’Etoile, France). The initial inoculum concentration was determined using the TEMPO system (BioMérieux, Marcy l’Etoile, France).

#### 2.1.3. Formation of Monospecies Biofilms In Vitro

For the formation of monospecies biofilms, 50 µL of the bacterial suspension were inoculated in the center of each coupon, resulting in an approximate concentration of 2.3 × 10^6^ CFU/cm^2^. The coupons were placed in sterile Petri dishes and subsequently introduced into a humid chamber and incubated at 3 °C to promote bacterial growth and subsequent biofilm formation [30]. The biofilms were kept incubated in these conditions for a week because, according to Ripolles-Avila et al. [31] this is the period required to generate in vitro mature biofilms (i.e., microbial cells disposed in multilayers with maximum biomass and water channels included) of *L. monocytogenes*. The maintenance of the biofilms was carried out by washing and renewing the nutrient medium. This process was performed in duplicate by washing the surfaces with 3 mL of sterile distilled water and providing them with 50 μL of new TSB culture medium. The objective of this protocol was to favor the development of adhered cells and to consequently consolidate mature biofilms. This maintenance protocol was carried out at two, three, six, and seven days of incubation, imitating what happens in the food industries where the surfaces are sanitized and worked on, thus providing them with new organic matter [31].

#### 2.1.4. Evaluation of Biofilm Formation through DEM

After a week of incubation, the stainless-steel coupons were washed twice with 3 mL of sterile distilled water to remove all the non-adhered cells. Subsequently, they were stained with 5 μL of the Live/Dead BacLight vital stain (Molecular Probes, Eugene, OR, USA) and protected from light at room temperature for 15 min. This kit uses two fluorochromic nucleic acids—SYTO 9 and propidium iodide. SYTO 9 can penetrate all cells, regardless of whether their membranes are damaged or intact. In contrast, the second fluorochrome (i.e., propidium iodide) only infiltrates the cells with an injured membrane, producing a reduction of the first dye applied (SYTO 9). Viable cells with an intact membrane show fluoresce in green and dead or injured cells in red. After incubation, the stained surfaces were evaluated by direct epifluorescent microscopy (DEM), using an Olympus BX51/BX52 direct epifluorescence microscope (Olympus, Tokyo, Japan) equipped with a 100 W mercury lamp (USH-103OL, Olympus) and a double-pass filter (U-M51004 F/R-V2, Olympus), and coupled to a digital camera (DP73, Olympus). The surfaces were observed with a 20× objective to verify that *B. safensis* could form biofilms. For each coupon, 10 images were taken from 10 different fields (Figure 1).

### 2.2. Evaluation of the Effect of Preformed B. safensis Biofilms on 22 Strains of L. monocytogenes

#### 2.2.1. *L. monocytogenes* Strains

Twenty-two strains of *L. monocytogenes* obtained as lyophilized cultures were used in this study (Table 1). Five of them came from the Spanish Type Culture Collection (CECT, University of Valencia, Valencia, Spain): CECT 5366, CECT 5672, CECT 5873, CECT 911, and CECT 935. Twelve were isolated from a pig processing industry in an earlier stage of study (A7, CDL69, EGD-e, P12, R6, S1 (R), S1 (S), S10 -1, S2-1, S2-2, and S2bac) by the National Institute for Agricultural and Food Research and Technology (INIA, Madrid, Spain). The other five strains were isolated from the surfaces of the same pig processing industry in a later phase of the study, as part of the national SURLIS project (RTA2014-00045-C03-03): Lm 1, Lm 2, Lm 3, Lm 4 and Lm 5.

Lyophilized strains were rehydrated in 9 mL of TSB and incubated at 30 °C for 40 to 48 h. After the incubation time, they were transferred to TSA plates and incubated at 37 °C for 18 to 24 h. The isolated colonies were further transferred to inclined TSA tubes. These were incubated at 37 °C for 24 h and then stored at a temperature of 4 °C, kept for a maximum period of one month.

#### 2.2.2. Inoculums Preparation

As indicated in Section 2.1.2, *B. safensis* inoculum was prepared according to the previous monospecies biofilm formation study. One week later, *L. monocytogenes* was inoculated on the surfaces where biofilms of *B. safensis* were previously formed (see Section 2.2.3). For this, a new culture of *L. monocytogenes* was made from the stock strains, which had been kept in inclined tubes, by streaking them on TSA plates. These plates were incubated at 37 °C for 18 to 24 h. The initial microbial concentration was estimated with the densitometer DENSIMAT (bioMérieux, Marcy l’Etoile, France). The concentration of the pathogen was estimated by means of a previous standard curve established between McFarland units and CFU/mL of *L. monocytogenes* [37]. The inoculum was prepared by transferring multiple isolated colonies in TSB until reaching a turbidity of 0.1 McFarland units, equivalent to approximately 10^6^ CFU/mL. The bacterial suspension was diluted decimally in 9 mL peptone water tubes. The initial concentration of the inoculum from the different strains of *L. monocytogenes* was determined using ALOA plates (bioMérieux, Marcy l’Etoile, France).

#### 2.2.3. In Vitro Formation of Monospecies Biofilms and Subsequent Implantation of *L. monocytogenes* on the Structure

For the formation of the biofilms, 50 µL of the prepared *B. safensis* suspension was inoculated in the center of each disc. The coupons were placed in sterile Petri dishes and incubated in a humid chamber at 30 °C. The biofilms were maintained for one week by washing and renewing nutrients at two, three, six, and seven days of incubation. The in vitro biofilm formation model was followed by a one week incubation period established by Ripolles-Avila et al. [31]. After the first seven days of incubation, 22 strains of *L. monocytogenes* were inoculated. The inoculum, consisting of 30 μL of *L. monocytogenes* bacterial suspension, was added in the center of all the preformed *B. safensis* biofilms. The *L. monocytogenes* controls were generated at the same time inoculating sterile coupons with 30 μL. The biofilms were maintained for two weeks with washing and periodic renewal of the nutrient medium [38,39]. This procedure was performed in duplicate by washing the coupons with 3 mL of sterile distilled water and transferring 50 μL of the TSB culture medium onto the coupons of mixed *B. safensis* biofilms and the controls, and 30 μL of TSB onto the biofilms of the control group of *L. monocytogenes*. The mixed biofilms and monospecies controls were incubated for another week under the same conditions and with the same frequency of washes and nutrient supply.

#### 2.2.4. Quantification of Biofilms by ALOA, TEMPO, and qPCR

Before each analysis (i.e., at two and seven days of incubation), the surfaces were washed in duplicate with 3 mL of sterile distilled water to discard the non-adhered cells, and then placed in sterile flasks containing 3.5 g of sterile glass beads and 9 mL of peptone water. Once the samples were introduced into the flasks, they were vortexed for 90 s at 40 Hz to detach the adhered cells from the surface for quantification. This model of cell detachment within biofilms has a high recovery efficiency of adhered microorganisms on the surface [16,30,37,38].

For the quantification in ALOA agar (BioMérieux, Marcy-l’Etoile, France) using the TEMPO system (BioMérieux, Marcy-l’Etoile, France), the resulting suspension was diluted decimally in 9 mL peptone water tubes. Next, 100 µL of the corresponding dilution was inoculated in ALOA plates and homogenized with a sterile Digralsky handle (Sudelab, Barcelona, Spain). For the TEMPO system, 1 mL of the sample was transferred to a vial previously hydrated with 3 mL of sterile distilled water. The vial was vortexed to homogenize its contents and transferred to the card through the TEMPO filling unit for subsequent incubation. The ALOA plates were incubated at 37 °C for 24 to 48 h and the TEMPO cards at 30 °C for 48 h. The necessary adjustments were then made to express the results in log (CFU/cm^2^).

The IQ-Check *L. monocytogenes* II PCR detection kit was used for the *L. monocytogenes* quantification by qPCR. This technique was applied as a complementary count to confirm the results obtained in ALOA. From the undiluted samples, 1.5 mL were transferred to different Eppendorf tubes (Eppendorf, Hamburg, Germany) and then centrifuged for 5 min at 12,000 rpm using the Heraeus Pico 17 centrifuge (Thermo Scientific, Madrid, Spain). The supernatant was discarded and 250 µL of the lysis reagent was added to the tubes, vortexed for 3 min, and placed in the heat block at 95 °C for 20 min. Subsequently, the tubes were centrifuged for 5 min at 12,000 rpm and 5 µL of the supernatant from the processed samples were introduced into the different strip wells for reaction and reading. Previously, 45 µL of the PCR mixture containing the amplifiers and fluorescent probes were introduced into the wells. The strips were sealed and placed into the MiniOpticon system (Biorad, Hercules, CA, USA), obtaining the results after 70 min. The Ct values obtained from each sample were analyzed using the CFX Manager software (Biorad, Hercules, CA, USA), and the respective calculations were made using a pre-established standard curve [37] to obtain the corresponding count values.

### 2.3. Evaluation of the Effect of Preformed Biofilms of Pseudomonas spp. and Candida zeylanoides on L. monocytogenes

The methodology used for both the preparation of the inoculum and the formation and quantification of the biofilms was the same as that described in Section 2.2. The only differences were in: i) the preformed biofilms, which were of *C. zeylanoides*, *Pseudomonas fluorescens*, and *Pseudomonas luteola*, isolated from the Iberian pig processing plant under study [14]; (ii) the strains of *L. monocytogenes*, which were Lm 2, Lm 3, and Lm 4 (i.e., code 19, 20, and 21), also isolated from the same processing plant [16]; and (iii) the quantification of the microorganisms, using only ALOA plates for *L. monocytogenes*, the TEMPO system for the total counts and the *Pseudomonas* spp. counts, and Sabouraud agar (Oxoid, Madrid, Spain) for the count of *C. zeylanoides*. These microorganisms are included to rule out the possible effects induced by the experimental design.

### 2.4. Characterization of the Effect of Preformed B. safensis Biofilms on L. monocytogenes

It was decided to analyze the water resulting from the sample washes to understand the effect of the preformed *B. safensis* biofilms. The methodology followed for the preparation of inoculums and the formation and quantification of biofilms was identical to that described in Section 2.2. The only modifications were: (i) the strains of *L. monocytogenes* used, which this time were Lm 2, Lm 3, Lm 4, and Lm 5 (i.e., code 19, 20, 21 and 22); (ii) the quantification of the microorganisms, using ALOA agar for *L. monocytogenes* and the TEMPO system for the total count of microorganisms; and (iii) the analysis of the washing water (not carried out in the previous studies), where 6 mL of wash water was recovered in sterile flasks, *L. monocytogenes* was quantified in ALOA plates, and the total number of microorganisms was evaluated using the TEMPO system.

### 2.5. Statistical Analysis

First, an exploratory analysis of the data was carried out and the Shapiro–Wilk test was performed to check the normality of the data sets (i.e., counts for each strain and time), using the statistical software package SPSS Statistics IBM 23 (Armonk, NY, USA). The bacterial counts obtained were converted to decimal logarithmic values to reduce the variability of the microbiological tests. The statistical software GraphPad Prism was used to analyze the results of each experiment. Given that all the data sets were compatible with a normal distribution (*p* > 0.05), a two-way ANOVA was run to evaluate each strain and time of analysis and to determine the differences between the counts of the mixed cultures and their respective controls for both *L. monocytogenes* and the rest of the microorganisms used, with a significance level of 5% (*p* < 0.05) for the analysis of all data. A posteriori contrast was performed using the Tukey test.

## 3. Results and Discussion

### 3.1. Evaluation of the Ability of B. safensis Biofilm Formation

It was considered essential to carry out this evaluation before continuing with the investigation since not all microorganisms can form well-organized biofilms. For example, *Campylobacter* spp. does not usually form its own biofilms and persists in food processing environments by invading biofilms formed by other bacteria [40]. In this case, as Figure 1 shows, *B. safensis* was able to form mature biofilms after the first week of incubation. The presence of viable cells (in green), injured cells (in red), and extracellular material (orange), in addition to a well-organized structure compatible with what has been observed in other studies [40,41] revealed the ability of *B. safensis* to form mature structures within this period of incubation. It has been suggested that the presence of interstitial voids are an indicator of biofilms in a mature state, representing water channels for the passage of nutrients and waste disposal [31]. According to the present results, these channels were observed for the formed *B. safensis* biofilms, revealing a complex and organized structure. This was consistent with our initial supposition, given that the microorganism was isolated from wild biofilms formed on Sensor Control Hygiene (SCH, Alinyma, Sant Cugat del Vallés, Spain) sensors located on different surfaces of the Iberian pig processing plant under study and was part of the dominant microbiota [14]. Therefore, it was demonstrated that the microorganism is capable of consolidating biofilms in a structured way.

### 3.2. Evaluation of the Effect of B. safensis Preformed Biofilms on 22 Strains of L. monocytogenes

Biofilms were analyzed at two and seven days of incubation to assess: (i) the implantation capacity of *L. monocytogenes*; and (ii) the final count of *L. monocytogenes* after one week of incubation, thereby determining if this pathogen could form mature biofilms under these conditions.

The average of the *L. monocytogenes* cell counts on ALOA agar from the extraction of the biofilms in the controls and the mixed biofilms is presented in Figure 2. In the presence of preformed biofilms of *B. safensis*, the tendency found was that the counts of all *L. monocytogenes* strains decreased significantly with respect to the monospecies biofilm controls at both two and seven days of incubation (Figure 2a,b). Statistically, the counts of *L. monocytogenes* after two days of incubation showed that 16 of the strains obtained a significantly lower load in mixed biofilms with *B. safensis* than in their controls (strain 1, *p* = 0.00; strain 2, *p* = 0.00; strain 3, *p* = 0.01; strain 5, *p* = 0.00; strain 8, *p* < 0.0001; strain 9, *p* = 0.00; strain 10, *p* < 0.0001; strain 11, *p* < 0.0001; strain 12, *p* = 0.00; strain 14, *p* = 0.00; strain 15, *p* = 0.01; strain 16, *p* < 0.0001; strain 17, *p* = 0.00; strain 18, *p* = 0.01; strain 20, *p* = 0.00; strain 21, *p* < 0.0001). Likewise, at seven days after inoculation of the pathogen, all the strains had a significantly lower log value (CFU/cm^2^) than their respective controls (strain 1, *p* = 0.00; strain 2, *p* = 0, 00; strain 3, *p* = 0.00; strain 4, *p* < 0.0001; strain 5, *p* = 0.00; strain 6, *p* = 0.00; strain 7, *p* < 0.0001; strain 8, *p* = 0.00; strain 9, *p* < 0.0001; strain 10, *p* < 0.0001; strain 11, *p* = 0.00; strain 12, *p* < 0.0001; strain 13, *p* = 0.00; strain 14, *p* = 0.00; strain 15, *p* < 0.0001; strain 16, *p* < 0.0001; strain 17, *p* <0.0001; strain 18, *p* < 0.0001; strain 19, *p* < 0.0001; strain 20, *p* = 0.00; strain 21, *p* < 0.0001; strain 22, *p* = 0.00). Figure 2a,b shows that a significant reduction was observed when the pathogen was inoculated on the preformed *B. safensis* biofilms. Nonetheless, the number of repetitions should be increased in subsequent studies to reduce the variability detected in the number of adhered cells in biofilms developed in the laboratory.

Multiple studies have evaluated the biocontrol capacity of *L. monocytogenes*, mainly through lactic acid bacteria such as *L. sakei*, which can produce bacteriocins [42]. For example, Winkelströter et al. [43] evaluated the formation of biofilms of *L. monocytogenes* in the presence of a strain of *L. sakei* and the cell-free supernatant of this microorganism containing sakacin. The presence of *L. sakei* and the bacteriocin supernatant inhibited the initial phases of biofilm formation of *L. monocytogenes,* observed by a decrease in the number of adhered cells present on the stainless-steel discs. However, a new growth of *L. monocytogenes* was observed in the culture containing sakacin after 48 h of incubation. Therefore, the inhibitory activity of this bacteriocin was not enough to exert definitive control of the pathogen under study. In a similar study, Pérez-Ibarreche et al. [44] investigated the effect of another bacteriocin producer strain of *L. sakei* on biofilms formed by *L. monocytogenes* on stainless-steel and polytetrafluoroethylene surfaces, materials commonly used in industries. This *L. sakei* strain was, in turn, effective in inhibiting biofilms. The authors suggested that prior treatment of food processing equipment with *L. sakei* or its bacteriocin could represent a possible method to prevent the adhesion of *L. monocytogenes* to surfaces. Thus, one of the possible explanations for the effect observed in the present study could be the ability of *B. safensis* to produce substances with an anti-*Listeria* or anti-adhesion effect. Along this line, further studies are required to determine if these kinds of substances are being produced by this microbial target.

Serotypes of *L. monocytogenes* that showed sensitivity to the presence of *B. safensis* were 1/2a, 1/2c and 4b (*p* < 0.05); serotypes 2a and 1/2b were affected, but not significantly (*p* = 0.33 and *p* = 0.23, respectively). It has been observed that the population structures generated in biofilms of *L. monocytogenes* by different serotypes can differ greatly [45]. To this effect, it is interesting to note that the results obtained at the serotype level could have a relevant application at the industrial level, since serotype 1/2a is the one most frequently isolated in food and environmental samples [7], and serotype 4b is related to the majority of listeriosis food-borne outbreaks [34].

*B. safensis* counts, obtained through the difference between the total count of microorganisms and the specific *L. monocytogenes* counts, did not show significant differences with the control (*p* > 0.05). Therefore, its load was not affected by the inoculation of the different strains of the pathogen in any of the evaluations. All this demonstrates how this microorganism can become constant over time once it generates a mature biofilm and constitutes its complex organization. Therefore, this could be one of the influential factors in the non-implantation of pathogens on industrial surfaces, revealing a promising field of study.

In line with the results obtained and those observed by other authors, two hypotheses were raised regarding the effect that the preformed *B. safensis* biofilms could have on *L. monocytogenes*: (i) non-specific competition hypothesis: the pre-established biofilms of other microorganisms affect the implantation of the pathogen on the surface by occupying the available space; and (ii) specific competition hypothesis: pre-existing microorganisms affect the development of the *L. monocytogenes* biofilm by competing for nutrients or producing anti-*Listeria* substances such as bacteriocins, anti-QS compounds, and enzymes, among others [43,46,47]. Other researchers have demonstrated these possible hypotheses; for example, *Bacillus subtilis* can produce enzymes that degrade the molecules used for the QS of *V. cholerae*, preventing its subsequent biofilms formation [46]. Leriche and Carpentier [34] also studied the effect of a mixed biofilm of *Staphylococcus sciuri* on *L. monocytogenes*, observing that the extracellular matrix of *S. sciuri* polysaccharides modified the balance between the planktonic phase and the biofilm of the pathogen, thus prevailing in the free form. The competition for nutrients associated with the anti-adhesive properties of *S. sciuri* could explain the decrease in *L. monocytogenes* counts. It has also been described that this microorganism has a high proteolytic activity, which can affect the *L. monocytogenes* matrix [39]. DNase and proteinase K are effective against biofilms of *L. monocytogenes*. Likewise, the addition of proteinase K completely inhibited the formation of biofilms [47].

These results call for the continued study of the mechanism of action of *B. safensis*. Previous study results have shown that the effect is not due to substances released into the medium since no real efficacy of cell-free filtered solutions was detected. Therefore, we should think about cellular components of *B. safensis* or substances released in the polymer matrix of this bacterium, which are only released in the biofilm and not in culture broths. In short, the use of microorganisms with the capacity to produce certain substances and/or compromise the survival and growth of certain pathogens would inhibit their long-term survival in facilities by preventing their generation and complete implantation in natural biofilm. This strategy would differ from those developed up until now, representing a potentially interesting ecological alternative tool to control pathogens in the food industry.

### 3.3. Evaluation of the Effect of Preformed Biofilms of Pseudomonas spp. and C. zeylanoides on L. monocytogenes

Considering the results obtained in the previous experiment, in which the behavior of 22 strains of *L. monocytogenes* against *B. safensis* was evaluated, it could not be said with certainty that this microorganism affected the formation of biofilms of *L. monocytogenes* given that the effect could have been induced by the experimental design itself. The lower counts of *L. monocytogenes* therefore needed to be shown to not be linked to (i) the charges of both microorganisms; (ii) the presence of a preformed biofilm of another microorganism, which would imply that *B. safensis* does not exert a specific anti-*Listeria* effect, but that this pathogen fails to overcome this ecological difficulty due to not having a great capacity for competition. For example, in a study carried out by Langsrud et al. [38], multiple bacteria that were part of the microbiota resident in the facilities of a salmon processing plant were isolated. A cocktail of several bacterial isolates from conveyor belts (*Listeria* spp., *Pseudomonas* spp., *Stenotrophomonas* spp., *Brochothrix* spp., *Serratia* spp., *Acinetobacter* spp., *Rhodococcus* spp. and *Chryseobacterium* spp.) was used to form biofilms on stainless-steel discs (12 °C, with salmon broth). After two days, they showed that *L. monocytogenes* represented 0.1 to 0.01% of the total population, while *Pseudomonas* spp. was the majority population [38]. As other authors have shown, by placing *L. monocytogenes* at a numerical disadvantage in terms of the initial load, this species usually shows a lower count without the other species present acting manifestly against it [48].

The same study was repeated to determine if there was real competition between *B. safensis* and *L. monocytogenes*, using three strains of the pathogen (belonging to the two serotypes most related to food and listeriosis, 1/2a and 4b) and three microorganisms, *C. zeylanoides*, *P. luteola,* and *P. fluorescens*, all isolated from the same facility. As indicated in the study conducted by Hascoët et al. [14], these microorganisms constitute an important part of the microbiota present on surfaces of the Iberian pig processing industry. Biofilms of the target microbiota started from a concentration of 4.00 × 10^5^ CFU/mL for *C. zeylanoides*; 2.15 × 10^6^ CFU/mL for *P. luteola*; 1.58 × 10^8^ CFU/mL for *P. fluorescens;* and 1.58 × 10^7^ CFU/mL as an average for *L. monocytogenes*. These concentrations were chosen following the criteria of other authors, who also inoculated in the range of 10^7^ CFU/mL of this pathogen in the formation of mixed biofilms [38,49].

No significant differences between these three organisms were observed regarding the counts of *L. monocytogenes* with respect to their controls (*p* > 0.05). Furthermore, as shown in Figure 3a–c, the trends observed were clearly different from those presented with *B. safensis*. To this effect, *L. monocytogenes* was not affected by the presence of a pre-existing biofilm on the surface of the three species evaluated. In the case of *C. zeylanoides* (Figure 3a), for strains 19 and 20, the counts decreased at two days and increased again at seven days. However, for strain 21, the counts decreased slightly after seven days. In this case, there appear to be no other published studies of the interaction between these two microorganisms.

With *P. luteola* (Figure 3b), the counts obtained from strain 19 were lower than the control, without being significant. For strains 20 and 21, the values were lower at two days and increased at seven days (+2 log (CFU/cm^2^) for strain 20). In the presence of *P. fluorescens* (Figure 3c), the total counts of strain 19 were slightly reduced, but its value increased at seven days, reaching higher levels than the control itself. Puga et al. [50] described that six days after their formation, biofilms reach (i.e., a certain state of maturity) a stage that not only implies cell dispersion, but also involves structural modifications and cell multiplication. This data could explain why the count values of *L. monocytogenes* increased again. A behavior similar to strain 19 was observed with strains 20 and 21, but levels close to those of the biofilm mono-species of *L. monocytogenes* were maintained. Puga et al. [51] also evaluated the effect of preformed biofilms of *P. fluorescens* on *L. monocytogenes*. In this case, they showed that the population of the pathogen was on average 1 to 2 logs (CFU/cm^2^) higher than when this microorganism grew in monospecies biofilms. It seems that *L. monocytogenes* can redesign the biofilm structure of *P. fluorescens* to favor its proliferation, compacting its structure. In fact, the arrival of the pathogen on the already established biofilms of *P. fluorescens* induced an overproduction of the extracellular matrix. These same authors also indicated that *L. monocytogenes* cells accumulated in the lower layers of pre-implanted biofilms, obtaining additional protection against possible physicochemical aggressions. A denser matrix can contribute to mixed biofilms being more resistant than monospecies biofilms to the external attack produced by enzymes, antimicrobial substances, or other agents [49]. The shrinkage of the matrix could be caused by the synthesis of an additional component of the extracellular matrix as a result of the interaction between species such as amyloid fibers [51]. This could explain, in part, why this pathogen can persist for long periods in the food industry environment [50].

In another study, the competitive capacity of different strains of *L. monocytogenes* was evaluated with Gram-negative and Gram-positive bacteria separately on stainless-steel surfaces at 12 °C. Mixed cultures included bacteria resident on the surfaces of salmon and meat processing industries such as *Acinetobacter* spp., *P. fragi*, *P. fluorescens*, *Serratia liquefaciens*, and *Stenotrophomonas maltophilia*. In this case, after nine days of incubation, the lowest counts of *L. monocytogenes* were found in the mixed biofilm with Gram-negative bacteria, dominated by *Pseudomonas* spp. [52]. All these results showed the existence of complex patterns in microbial interactions. Thus, the inhibition of *L. monocytogenes* in multi-species cultures depends not only on cell contact but on various mechanisms involved in tolerance and antagonism between bacterial species.

### 3.4. Characterization of the Effect of Preformed Biofilms of B. safensis on L. monocytogenes

The results of the evaluation of the effect of the preset biofilms of *C. zeylanoides*, *P. luteola*, and *P. fluorescens* confirmed that *B. safensis* had a negative impact on the formation of *L. monocytogenes* biofilms. The next question focused on how this microorganism affected the pathogen, although to a lesser extent. To test the hypotheses presented in Section 3.2, the washing water analysis was carried out with the objective of better characterizing the effect of these preformed biofilms. In fact, given the results obtained with *B. safensis*, it was thought that, for some reason, the biofilm of this microorganism did not allow *L. monocytogenes* to become established on the surfaces. Thus, all *L. monocytogenes* cells that had not adhered to the surface would have been washed away by the washing water.

The newly formed biofilms for this experiment were generated from 1.04 × 10^6^ CFU/mL inoculums of *B. safensis* and an average of 1.03 × 10^6^ CFU/mL of the *L. monocytogenes* strains. The results of the washing water analysis did not confirm the initial hypothesis. The trend, shown in Figure 4, indicated that the load of *L. monocytogenes* in the washing water of the mixed biofilms was lower than the load of the washing water of the control biofilms of *B. safensis*. In this regard, all the strains showed a significantly lower count (*p* < 0.0001). Three strains of the four tested showed significant differences at seven days (strain 19, *p* = 0.00; strain 20, *p* < 0.0001; strain 21, *p* < 0.0001).

The observed trend indicates that *B. safensis* affected the viability and implantation of the pathogen, since after two days, *L. monocytogenes* could not be implanted correctly and was not recovered in the washing water. Furthermore, at seven days, the counts remained significantly lower (*p* < 0.05) than those obtained from the control biofilms. These results indicate that *B. safensis* not only competed for space but also affected the pathogen’s viability. This fact again reveals the possibilities that the use of this microorganism at the industrial level could have been because of its profile for ecological replacement and its ability to produce certain substances that affect a pathogen as important as *L. monocytogenes*. However, more studies would be needed to determine its effectiveness in real conditions and thus be able to specify its use as a joint tool in the control of biofilms.

## 4. Conclusions

The preformed mature biofilms of *B. safensis* can inhibit the formation of biofilms of multiple strains of *L. monocytogenes*, destroying the pathogen by means of a currently unspecified mechanism. Further studies are required to know the mechanism(s) responsible for this effect, but what is now known is that this is not only due to a simple competition for space or nutrients. Given the non-enterotoxigenic properties of *B. safensis*, favoring the establishment of this microorganism on certain surfaces of the meat industries not directly in contact with food such as sinks and other persistence niches, this could represent one more way to combat the persistence of *L. monocytogenes*, together with other bacterial inhibitors and proper hygiene operations. Complementary studies are required to evaluate the interaction of *B. safensis* with other relevant foodborne pathogens.

## Figures and Tables

**Figure 1 biomolecules-11-00475-f001:**
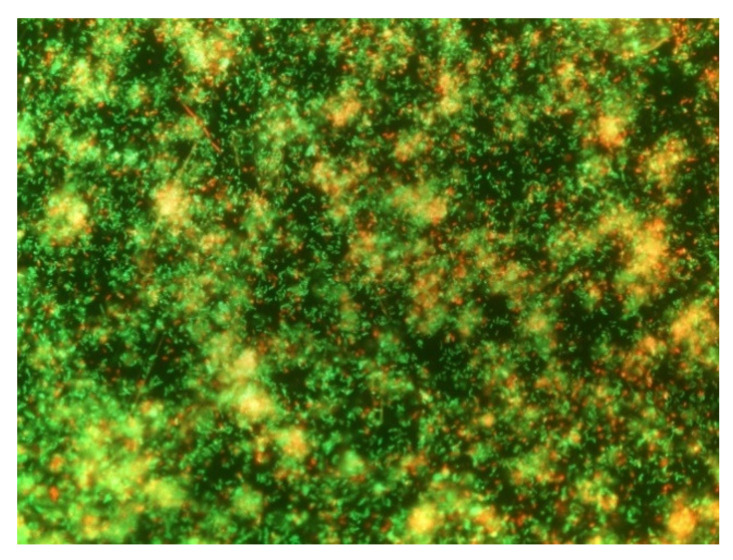
Representative image of *B. safensis* biofilms formed after one week of incubation, observed with Live/Dead BacLight staining by direct epifluorescence microscopy (20× objective).

**Figure 2 biomolecules-11-00475-f002:**
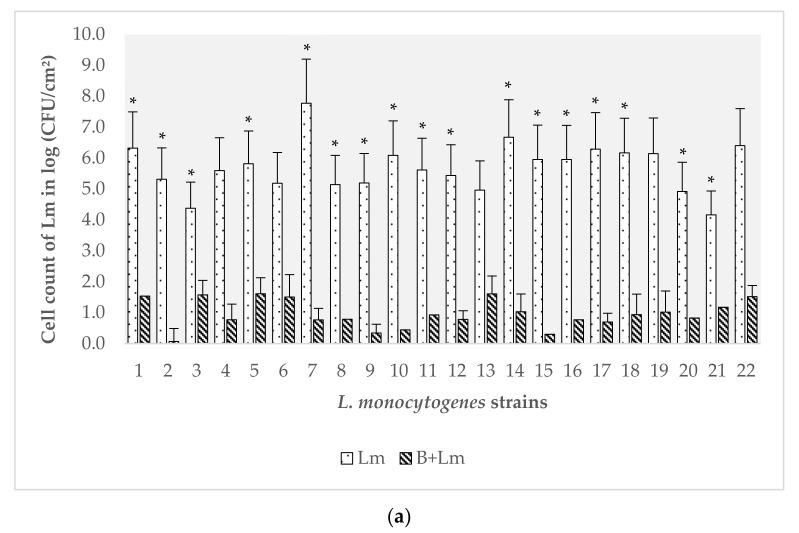
Counts on ALOA agar of the 22 *Listeria monocytogenes* strains as controls (Lm), and the respective counts in the mixed biofilms with *B. safensis* (B + Lm) at two days (**a**) and seven days (**b**) of incubation. The error bars represent the standard error of the mean (n = 6). * indicates significant differences (*p* < 0.05) between the experimental group (B + Lm) and its respective control (Lm).

**Figure 3 biomolecules-11-00475-f003:**
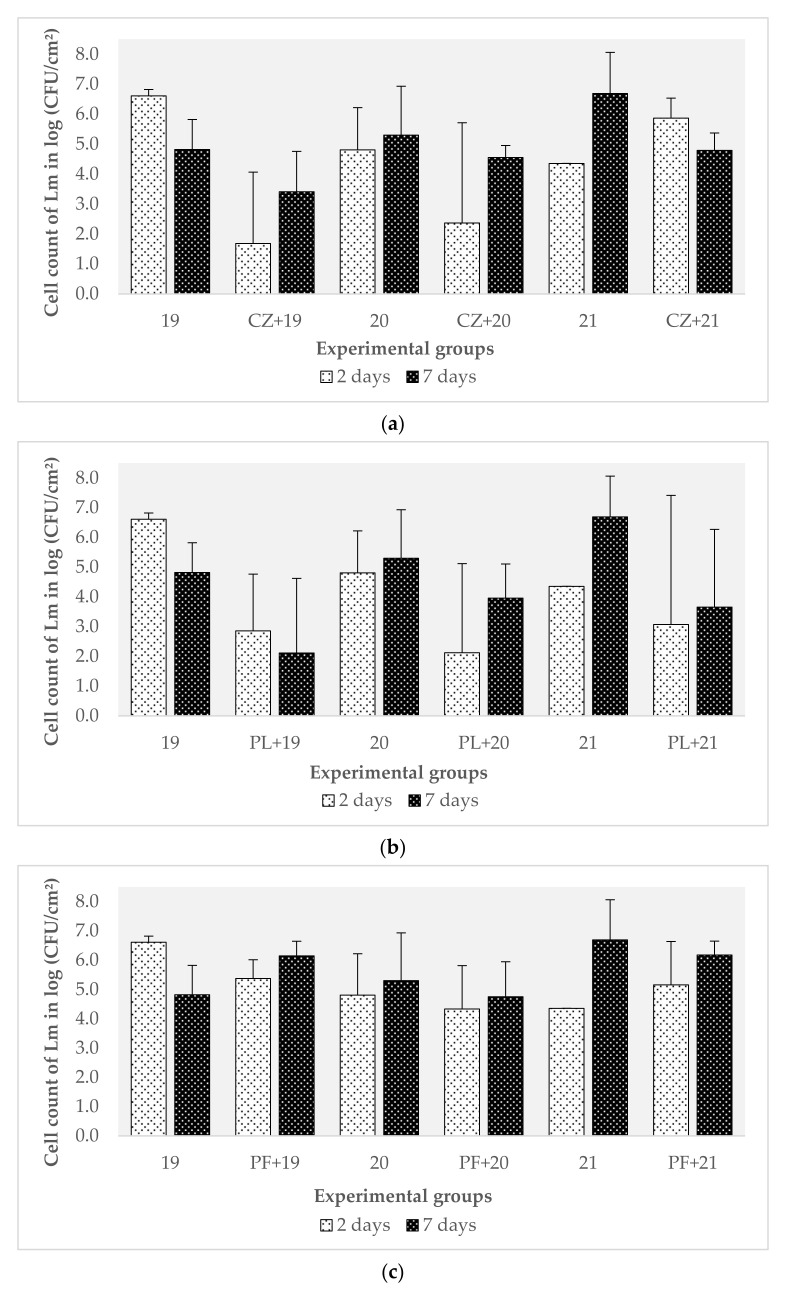
Counts on ALOA agar of *Listeria monocytogenes* strains from monospecies biofilm controls (Lm 19, 20, 21) and mixed biofilms with: (**a**) *Candida zeylanoides*; (**b**) *Pseudomonas luteola*; and (**c**) *Pseudomonas fluorescens,* at two and seven days of incubation. The error bars represent the standard error of the mean (n = 4).

**Figure 4 biomolecules-11-00475-f004:**
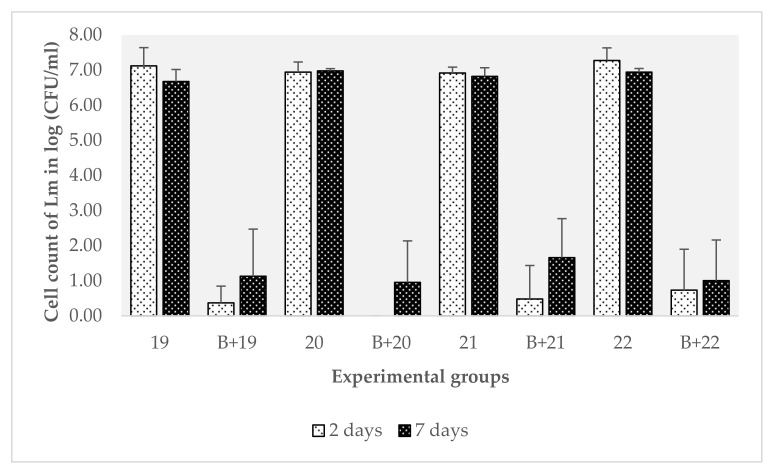
Counts on ALOA agar of *Listeria monocytogenes* strains from the washing solutions of control biofilms (Lm 19, 20, 21, 22) and mixed biofilms with *B. safensis* at two and seven days of incubation. The error bars represent the standard error of the mean (n = 4). For the Lm 3 strain for which no two-day count was observed (B + 20), the mean count of *L. monocytogenes* was 0 CFU/mL.

**Table 1 biomolecules-11-00475-t001:** *Listeria monocytogenes* strains used in the studies.

Code	Strain	Serotype	Origin
1	4423	1/2a	[32]
2	5873	1/2a	CECT
3	A7	1/2a	[33]
4	CDL69	1/2a	[32]
5	EGD-e	1/2a	[32]
6	P12	1/2a	[34]
7	R6	1/2a	[35]
8	S1(R)	1/2a	[36]
9	S1(S)	1/2a	[36]
10	S2-1	1/2a	[36]
11	S2-2	1/2a	[32]
12	S2^bac^	1/2a	[36]
13	5366	4b	CECT
14	5672	4b	CECT
15	935	4b	CECT
16	S10-1	2a	[32]
17	911	1/2c	CECT
18	Lm 1	4b	Isolated from an industrial meat processing environment in 2017 [16]
19	Lm 2	4b	
20	Lm 3	1/2a	
21	Lm 4	1/2a	
22	Lm 5	1/2b	

CECT—Spanish Type Culture Collection.

## Data Availability

The data presented in this study are available on request from the corresponding author. The data are not publicly available due to research group policy.

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
