# Peer review of "In Vitro Preformed Biofilms of Bacillus safensis Inhibit the Adhesion and Subsequent Development of Listeria monocytogenes on Stainless-Steel Surfaces"

_biomolecules, 2021, doi:10.3390/biom11030475_

Round 1
Reviewer 1 Report
The results of the present work show that B. safensiscould be a promising strategy to combat L. monocytogenes biofilms at the industrial level; although, as underlined by the authors further research should be performed.
Some minor revisions must be resolved.
SPECIFIC COMMENTS:
Introduction
Bacillus safensis is barely introduced in this section. I would suggest including a description of this bacterium to understand the reason that led to choose it for the experiment.
Line 31: Literature cited is not up to date. There are more recent EFSA opinion on it.
Moreover it could be useful to include the description of other strategies for combating biofilm, so please, in lines 56-57 insert the AMPs as innovative and effective strategy to reduce Listeria monocytogenes growth and biofilms production.
I would suggest these references: Palmieri G, Balestrieri M, Capuano F, Proroga YTR, Pomilio F, Centorame P, Riccio A, Marrone R, Anastasio A. Bactericidal and antibiofilm activity of bactenecin-derivative peptides against the food-pathogen Listeria monocytogenes: New perspectives for food processing industry (2018) Int. J. Food Microbiol. 279, 33.
Materials and Methods
Line 127: Please, replace “was” with “were”.
Results & Discussion
Figure 2 caption: I can imagine that figure 2a is the counts of the 22 Listeria monocytogenes strains at 2 days of incubation and Figure 2b at 7 days. Please indicate it clearly in the caption.
Lines 372-387: I would suggest including (summing up) these information in the introduction. See above comment.
Lines 430-434: “ Multiple bacteria that were part of the microbiota resident in the facilities of a salmon processing plant were isolated. A cocktail of several bacterial isolates from conveyor belts (Listeria spp., Pseudomonas spp., Stenotrophomonas spp., Brochothrix spp., Serratia spp., Acinetobacter spp., Rhodococcus spp. and Chryseobacterium spp.) was used to form biofilms on stainless steel discs” in which study? These information seem to regard the present work and not a previous one performed by other authors. Please explain better.
Author Response
Reviewer #1:
The results of the present work show that B. safensis could be a promising strategy to combat L. monocytogenes biofilms at the industrial level; although, as underlined by the authors further research should be performed.
Some minor revisions must be resolved.
SPECIFIC COMMENTS:
Introduction
Bacillus safensis is barely introduced in this section. I would suggest including a description of this bacterium to understand the reason that led to choose it for the experiment.
Response:
Thank you very much, we tried to improve it through an extensive description (see lines 85-105).
Line 31: Literature cited is not up to date. There are more recent EFSA opinion on it.
Response:
Many thanks. We have updated the data and the reference employed (see lines 31-32).
Moreover it could be useful to include the description of other strategies for combating biofilm, so please, in lines 56-57 insert the AMPs as innovative and effective strategy to reduce Listeria monocytogenes growth and biofilms production.
Response:
This information has been included as suggested.
I would suggest these references: Palmieri G, Balestrieri M, Capuano F, Proroga YTR, Pomilio F, Centorame P, Riccio A, Marrone R, Anastasio A. Bactericidal and antibiofilm activity of bactenecin-derivative peptides against the food-pathogen Listeria monocytogenes: New perspectives for food processing industry (2018) Int. J. Food Microbiol. 279, 33.
Response:
Thank you for your suggestion, we have added it to the text (see lines 57-58).
Materials and Methods
Line 127: Please, replace “was” with “were”.
Response:
Sorry about that. We have corrected it (see line 143).
Results & Discussion
Figure 2 caption: I can imagine that figure 2a is the counts of the 22 Listeria monocytogenes strains at 2 days of incubation and Figure 2b at 7 days. Please indicate it clearly in the caption.
Response:
Totally agree, many thanks. We have addressed this suggestion (see the caption of Figure 2, line 348).
Lines 372-387: I would suggest including (summing up) these information in the introduction. See above comment.
Response:
Thank you. We have tried to improve it by adding more information on the instruction section (see lines 89-102).
Lines 430-434: “ Multiple bacteria that were part of the microbiota resident in the facilities of a salmon processing plant were isolated. A cocktail of several bacterial isolates from conveyor belts (Listeria spp., Pseudomonas spp., Stenotrophomonas spp., Brochothrix spp., Serratia spp., Acinetobacter spp., Rhodococcus spp. and Chryseobacterium spp.) was used to form biofilms on stainless steel discs” in which study? These information seem to regard the present work and not a previous one performed by other authors. Please explain better.
Response:
Thanks for your observation. We have corrected it (see lines 456-462).
Reviewer 2 Report
The authors describe the proof of concept that a "positive" biofilm of Bacillus safensis could prevent the development of Listeria monocytogenes on plant surfaces. The in vitro experiments show that i) B. safensis is able to develop a mature biofilm and ii) the B. safensis biofilm decrease the adhesion and the viability of 22 strains of L. monocytogenes.
The aim of this article is interesting and could be a good strategy to limit L. monocytogenes in food industries. The inhibition results were clear but they would be better if experiments were be performed in triplicate. I also wondered if the effect of Pseudomonas and Candida biofilms was really relevant in this article. The molecules potentially involved in the mechanism were only discussed and could have been deeper investigated to submit in a journal on biomolecules.
Some comments:
-it would be interesting to describle B. safensis species in the introduction
-there is a lot of repetitions in the material and methods and some paragraphs could be merged.
Figure 2: indicate a) and b) meanings
Figure 3: Switch the bars to have control close to result by Listeria strain
Author Response
Reviewer #2:
The authors describe the proof of concept that a "positive" biofilm of Bacillus safensis could prevent the development of Listeria monocytogenes on plant surfaces. The in vitro experiments show that i) B. safensis is able to develop a mature biofilm and ii) the B. safensis biofilm decrease the adhesion and the viability of 22 strains of L. monocytogenes.
The aim of this article is interesting and could be a good strategy to limit L. monocytogenes in food industries. The inhibition results were clear but they would be better if experiments were be performed in triplicate. I also wondered if the effect of Pseudomonas and Candida biofilms was really relevant in this article. The molecules potentially involved in the mechanism were only discussed and could have been deeper investigated to submit in a journal on biomolecules.
Response:
Thank you for your comment. We included these additional microorganisms to show that the results obtained with Bacillus safensis were not caused by the experimental design. We wanted to confirm that preformed biofilms by other microbial species in these conditions do not affect Listeria monocytogenes. This has been included in the text to reinforce the reason why these microorganisms were used.
Furthermore, regarding the inhibition results which are the most relevant results due to these manifest the potential application, they were be performed in triplicate (duplicates in three separate days; n=6). The experiments that were done in duplicates in two separate days were the ones that were performed to check the experimental conditions (influence of Pseudomonas and Candida biofilms).
Finally, as for the mechanism of action of B. safensis, we are continuing to investigate and hope to be able to explain it in another publication soon.
Some comments:
-it would be interesting to describe B. safensis species in the introduction.
Response:
We corrected it, many thanks.
-there is a lot of repetitions in the material and methods and some paragraphs could be merged.
Response:
Thank you for the observation, we appreciate it. We have tried to organize the manuscript so that the experiments and the experimental design is fully understandable and replicable by other researchers. For the moment, we have tried to summarize some parts. If editor also considers that, we will try to merge it more.
Figure 2: indicate a) and b) meanings.
Figure 3: Switch the bars to have control close to result by Listeria strain.
Response:
Done, many thanks.
Reviewer 3 Report
- What do you call preformed mature biofilms of B. safensis? Needs to be explained.
- Eliminating the pathogen through a currently unidentified mechanism, ... if you do not know the mechanism, how do you determine that it is an elimination and not a bacteriostatic effect?
- I think Bacillus safensis should be in your keywords.
- How did you determine that 2.3 x 128 106 CFU / cm2 was the most suitable concentration to make the biofilm? I think that needs to be explained.
- In page 3: The objective of this protocol was to favor the 136 development of adhered cells and to consequently consolidate mature biofilms. This 137 maintenance protocol was carried out at 2, 3, 6, and 7 days of incubation, imitating what 138 happens in the food industries where the surfaces are sanitized and worked on, thus 139 providing them with new organic matter [26]. It would not be comparable since, as you say in the industry, sanitizer is used and in this case you are only rinsing with water, or is it that sanitizers have no effect whatsoever?
- Page 4. The surfaces were observed with a 20X objective to verify that B. safensis could form biofilms. For each coupon, ten images were taken from 10 different fields. Indicate here that the images are shown later
- The methodology is repeated a bit and some things like those mentioned in the previous points (4.6) are not made explicit
- Page 5: The sterile coupons were in-202 oculated at the same time, also with 30 μl of the initial bacterial suspensions, which bacterial suspension are you referring to? The wording of point 2.2.3. it's confusing
- In point 2.3., It would be desirable to explain why these strains were also used.
- Page 9 anti Listeria effect or anti-adhesion effect?
Author Response
Reviewer #3:
- What do you call preformed mature biofilms of B. safensis? Needs to be explained. Response:
Understood, we have included an extended description to highlight the main characteristics (see lines 149-150).
- Eliminating the pathogen through a currently unidentified mechanism, ... if you do not know the mechanism, how do you determine that it is an elimination and not a bacteriostatic effect?
Response:
Thank you for your comment. We consider that there is an elimination due to the great reduction in Listeria monocytogenes counts, both in the biofilms and in the wash water. Furthermore, in other tests that we are currently performing in our lab, for other L. monocytogenes strains, we found that not growth after 7 days is observed. In order to check if the population is sublethaly damaged and therefore the effect could be bacteriostatic, we have performed ISO 11290 part 1. In those cases, we have not obtained any growth. This reinforces our finding regarding the bactericidal effect.
- I think Bacillus safensis should be in your keywords.
Response:
Done, thank you (see line 26).
- How did you determine that 2.3 x 128 106 CFU / cm2 was the most suitable concentration to make the biofilm? I think that needs to be explained.
Response:
Thanks for your observation. We decided to start from a large concentration to be able to see the complete reduction in logarithms. Starting from this microbial level led us to observe more clearly the effect and differences.
- In page 3: The objective of this protocol was to favor the development of adhered cells and to consequently consolidate mature biofilms. This maintenance protocol was carried out at 2, 3, 6, and 7 days of incubation, imitating what happens in the food industries where the surfaces are sanitized and worked on, thus providing them with new organic matter [26]. It would not be comparable since, as you say in the industry, sanitizer is used and in this case you are only rinsing with water, or is it that sanitizers have no effect whatsoever?
Response:
Thank you very much, we understand your comment and we appreciate it. However, we consider that, although disinfectants are used in the food industry daily and obviously these have an effect on microorganisms, the main treatment that helps to break the biofilm matrix is ​​an adequate cleaning treatment (this implies that if surfaces are not highly cleaned, biofilms will remain independently that biocides are applied). Under our perspective, there are many places in the food industry where perfect cleaning and disinfection is not always carried out. For example, in corners, sinks, among others…, an ideal concentration of detergent and disinfectant is not always reached. In that case, the procedure followed through this study can “mimic” industrial reality or produce more representative results. In any case, this procedure has been proposed with the aim of consolidating mature biofilms, because these structures in this stage (mature) are the most problematic.
- Page 4. The surfaces were observed with a 20X objective to verify that B. safensis could form biofilms. For each coupon, ten images were taken from 10 different fields. Indicate here that the images are shown later.
Response:
Thanks. We have corrected it (line 174).
- The methodology is repeated a bit and some things like those mentioned in the previous points (4.6) are not made explicit.
Response:
Thank you for the observation, we appreciate it. We have tried to organize the manuscript so that the experiments and the experimental design is fully understandable and replicable by other researchers. For the moment, we have tried to summarize some parts. If editor also considers that, we will try to merge it more.
- Page 5: The sterile coupons were inoculated at the same time, also with 30 μl of the initial bacterial suspensions, which bacterial suspension are you referring to? The wording of point 2.2.3. it's confusing.
Response:
We have addressed this issue. Now, you can find this part re-worded in Line 219.
- In point 2.3., It would be desirable to explain why these strains were also used.
Response:
We have included a sentence to explain the reason why these strains have been used. This information can be found in lines 272-273.
- Page 9 anti Listeria effect or anti-adhesion effect?
Response:
Thank you. Both of the effects are correct. We have included “anti-adhesion effect” in the text (see line 397).
Reviewer 4 Report
The research presented are interesting and extend the knowledge about antagonistic behavior of the microorganisms. However, I have some comments and questions, which need to be addressed before the acceptance of the text to be published in Biomolecules.
Major comments:
- General: Antagonistic effect between different microorganisms growing in the same niche is commonly observed, and humans try to exploit this phenomenon to fight unfavorable microbes. However, food processing plants are quite specific enterprises and except of the microorganisms used in biotechnological process of food production (e.g. lactic acid bacteria), the halls and equipment should have as low microbial load as possible. Can the purpose introduction of additional microorganism, even nonpathogenic as safensis, be dangerous? Moreover, B. safensis produces a great number of enzymes and secondary metabolites, which can change food properties by e.g. food fermentation or enzymatic degradation [DOI: 10.1515/biolog-2015-0062]. How to reconcile it?
- Introduction: Give more recent epidemiological data on monocytogenes infections / outbreaks.
- Introduction (line 33-34): “listeriosis presents the highest proportion of cases needing hospitalization among all the zoonoses under surveillance” – The sentence is doubly wrong: first of all listeriosis is food-borne disease, not zoonosis. Secondly most cases of listeriosis run as asymptomatic or mild infection. While monitored real zoonotic disease such as anthrax usually occurs as very severe. Have to be improved.
- Introduction (line 90-94): What is the difference between specific objective (ii) “evaluation of the effect of preformed biofilms of safensis on 22 strains of L. monocytogenes” and specific objective (iv) “characterization of the effect of the biofilms of B. safensis on L. monocytogenes”? Improve the description to be more clear.
- Materials and Methods (subsection 2.1.1.): Why did you use double disinfection procedure for the materials: first autoclaving, then 70% isopropanol?
- Materials and Methods (subsection 2.1.3.): Did you include negative control (material treated with TSB alone) to check material sterility and correctness of the experience?
- Materials and Methods (subsection 2.1.4.): It seems to be difficult to stain the materials 2 cm in diameter with 5µl LIVE/DEAD dyes. Precise description.
- Materials and Methods (subsection 2.2.4.) and Results: “The IQ-Check monocytogenes II PCR detection kit was used for the L. monocytogenes quantification” – When exactly this method were used? Indicate which results were calculated based on qPCR – see also next comment.
- Figures descriptions: Add the method used for microbial cell counting.
- Results and Discussion (line 292-293): “According to the present results, these channels were observed for the formed safensis biofilms…” – How did you observe the channels in fluorescent microscope giving “one-layer” picture (from the top)?
- Results and Discussion (line 385-386): “Another interesting finding is that safensis is capable of blocking virulence factors and the formation of biofilms of Candida neoformans …” – The is no “Candida neoformans” – The name of this pathogenic fungi is Cryptococcus neoformans
- Results and Discussion (line 562): “The observed trend indicates that safensis affected the viability and implantation…” – Based on your results B. safensis affected L. monocytogenes cell viability and division rather, than implantation. If cell adhesion to the materials would be blocked, you should recovery almost all L. monocytogenes inoculum in the washing water (log=6). Since microbial adhesion starts very quickly (2-4 h), you may prepare in future such short-lasting experiment to clarify B. safensis effect on L. monocytogenes adhesion (implantation).
Minor comments:
- Introduction (line 83-86): “Iberian pig processing plant…” – The sentence is too long and complicated. Improve.
- Materials and Methods (subsection 2.1.1.): Complete a source of Grade 2B AISI 316 stainless steel coupons.
- Fig. 2. description: “(i.e., monospecies biofilms)” – Monospecies L. monocytogenes biofilms are the only controls, so delete “e.g.”
- Fig. 2. description: “* indicates significant differences (P < 0.05) between the experimental group and its respective control” – For clarity complete the description by pointing to appropriate groups abbreviations in the brackets, e.g. control (Lm)
Author Response
Reviewer #4:
The research presented are interesting and extend the knowledge about antagonistic behavior of the microorganisms. However, I have some comments and questions, which need to be addressed before the acceptance of the text to be published in Biomolecules.
Major comments:
- General: Antagonistic effect between different microorganisms growing in the same niche is commonly observed, and humans try to exploit this phenomenon to fight unfavorable microbes. However, food processing plants are quite specific enterprises and except of the microorganisms used in biotechnological process of food production (e.g. lactic acid bacteria), the halls and equipment should have as low microbial load as possible. Can the purpose introduction of additional microorganism, even nonpathogenic as safensis, be dangerous? Moreover, B. safensis produces a great number of enzymes and secondary metabolites, which can change food properties by e.g. food fermentation or enzymatic degradation [DOI: 10.1515/biolog-2015-0062]. How to reconcile it?
Response:
Thank you for your observation. The general idea is not really to introduce B. safensis, but as it was isolated from the plant's own facilities, to use it more as a safety marker and / or to favor its presence with respect to other microorganisms that favor the persistence of Listeria monocytogenes.
We did other (unpublished) studies that showed that B. safensis did not affect the products made in that Iberian pig processing plant.
We understand that it is something "daring" but the idea is that these types of industries will never achieve sterile surfaces, the idea is then to monitor and modulate the microbiota (even if it is minimal).
- Introduction: Give more recent epidemiological data on monocytogenes infections / outbreaks.
Response:
Thank you. We tried to improve it (see lines 31-32).
- Introduction (line 33-34): “listeriosis presents the highest proportion of cases needing hospitalization among all the zoonoses under surveillance” – The sentence is doubly wrong: first of all listeriosis is food-borne disease, not zoonosis. Secondly most cases of listeriosis run as asymptomatic or mild infection. While monitored real zoonotic disease such as anthrax usually occurs as very severe. Have to be improved.
Response:
Many thanks. We also have corrected it (see lines 34-35).
- Introduction (line 90-94): What is the difference between specific objective (ii) “evaluation of the effect of preformed biofilms of safensis on 22 strains of L. monocytogenes” and specific objective (iv) “characterization of the effect of the biofilms of B. safensis on L. monocytogenes”? Improve the description to be more clear.
Response:
The difference within both objectives is the target assay performed on the specific parameter that was analyzed and the reason why it was tested (i.e. biofilm preformed on surface or the detachment of L. monocytogenes cells in the washing water solution). That is why we differentiated both objectives. We started with the second one (ii), and then, when we were observing the effect, we wanted to know if we have a constant detachment (that is why we did the iv objective).
- Materials and Methods (subsection 2.1.1.): Why did you use double disinfection procedure for the materials: first autoclaving, then 70% isopropanol?
Response:
First, we wash with detergent and then we apply isopropanol. Once this is finished, we autoclave the surfaces so that the discs are completely sterile. We are following the European standard UNE-EN 13697:2015 as mentioned line 122. We hope that now is clearer.
- Materials and Methods (subsection 2.1.3.): Did you include negative control (material treated with TSB alone) to check material sterility and correctness of the experience?
Response:
These controls are not integrated into the experiment but are part of our routine in the laboratory, we carry out controls to each batch of preparation of sterile material.
- Materials and Methods (subsection 2.1.4.): It seems to be difficult to stain the materials 2 cm in diameter with 5µl LIVE/DEAD dyes. Precise description.
Response:
We do this protocol after years of trying the best combinations and we also put a coverslip so that all the drop is well distributed. Putting more staining solution is not ideal because it leaks and we have more difficulties to observe the biofilms more nitid in the microscope. You can review the protocol in the following publication: C. Ripolles-Avila, A.S. Hascoët, A.E. Guerrero-Navarro, J.J. RodríguezJerez, Establishment of incubation conditions to optimize the in vitro formation of mature Listeria monocytogenes biofilms on food-contact surfaces, Food Control (2018), doi: 10.1016/j.foodcont.2018.04.054
- Materials and Methods (subsection 2.2.4.) and Results: “The IQ-Check monocytogenes II PCR detection kit was used for the L. monocytogenes quantification” – When exactly this method were used? Indicate which results were calculated based on qPCR – see also next comment.
Response:
Thanks for your observation. We have added some more explanation (see line 247).
- Figures descriptions: Add the method used for microbial cell counting.
Response:
Many thanks. We have correct it.
- Results and Discussion (line 292-293): “According to the present results, these channels were observed for the formed safensis biofilms…” – How did you observe the channels in fluorescent microscope giving “one-layer” picture (from the top)?
Response:
Water channels have been highlighted to be an indication of biofilm maturity. It has been hypothesised that the void areas may represent water channels that promote nutrient circulation and waste removal (Costerton, Lewandowski, Caldwell, Korber, & Lappin-Scott, 1995; Donlan & Costerton, 2002). In a one-layer picture, we can observe it from the top as empty voids. L. monocytogenes biofilms are normally organised into honeycomb-like structures, as well as other microorganisms. Within this structure, void areas are observed. It is very visual. In other studies, we have used CLSM and results are completely in line with DEM observations (see publication: C. Ripolles-Avila, A.S. Hascoët, A.E. Guerrero-Navarro, J.J. RodríguezJerez, Establishment of incubation conditions to optimize the in vitro formation of mature Listeria monocytogenes biofilms on food-contact surfaces, Food Control (2018), doi: 10.1016/j.foodcont.2018.04.054).
- Results and Discussion (line 385-386): “Another interesting finding is that safensis is capable of blocking virulence factors and the formation of biofilms of Candida neoformans …” – The is no “Candida neoformans” – The name of this pathogenic fungi is Cryptococcus neoformans
Response:
We are sorry for the mistake, many thanks, we have correct it (see line 101).
- Results and Discussion (line 562): “The observed trend indicates that safensis affected the viability and implantation…” – Based on your results B. safensis affected L. monocytogenes cell viability and division rather, than implantation. If cell adhesion to the materials would be blocked, you should recovery almost all L. monocytogenes inoculum in the washing water (log=6). Since microbial adhesion starts very quickly (2-4 h), you may prepare in future such short-lasting experiment to clarify B. safensis effect on L. monocytogenes adhesion (implantation).
Response:
Thanks for your recommendation.
Minor comments:
- Introduction (line 83-86): “Iberian pig processing plant…” – The sentence is too long and complicated. Improve.
Response:
Understood. We have tried to improve it (see lines 85-88).
- Materials and Methods (subsection 2.1.1.): Complete a source of Grade 2B AISI 316 stainless steel coupons.
Response:
We have completed, many thanks (see line 116).
- Fig. 2. description: “(i.e., monospecies biofilms)” – Monospecies L. monocytogenes biofilms are the only controls, so delete “e.g.”
Response:
Thank you, we have corrected it (see line 348).
- Fig. 2. description: “* indicates significant differences (P < 0.05) between the experimental group and its respective control” – For clarity complete the description by pointing to appropriate groups abbreviations in the brackets, e.g. control (Lm)
Response:
We tried to improve it (see line 378). Thank you.
Round 2
Reviewer 2 Report
Thank you for your responses. All modifications improve the manuscript.